# Integrating full and partial genome sequences to decipher the global spread of canine rabies virus

Andrew Holtz [1] ✉, Guy Baele [2], Hervé Bourhy [1,3] & Anna Zhukova [4] ✉

Despite the rapid growth in viral genome sequencing, statistical methods face challenges in handling historical viral endemic diseases with large amounts of underutilized partial sequence data. We propose a phylogenetic pipeline that harnesses both full and partial viral genome sequences to investigate historical pathogen spread between countries. Its application to rabies virus (RABV) yields precise dating and confident estimates of its geographic dispersal. By using full genomes and partial sequences, we reduce both geographic and genetic biases that often hinder studies that focus on specific genes. Our pipeline reveals an emergence of the present canine-mediated RABV between years 1301 and 1403 and reveals regional introductions over a 700-year period. This geographic reconstruction enables us to locate episodes of human-mediated introductions of RABV and examine the role that European colonization played in its spread. Our approach enables phylogeographic analysis of large and genetically diverse data sets for many viral pathogens.

Studies that investigate the dynamics of viral emergence are vital to inform epidemiological decision making[1–3]. Whole-genome sequencing has become the norm for phylogenetic studies focused on modern epidemics/pandemics, such as the recent SARS-CoV-2 pandemic[4]. Alternatively, many zoonotic pathogens have decades of partial genome submissions, greatly outnumbering the number of whole-genome sequences (WGS)[5]. Such neglected zoonotic diseases—such as those caused by West Nile, rabies, and Lassa viruses—receive inadequate attention from public health officials, leading to limited funding[6] and a focus on sequencing only certain parts of the viral genome, particularly in cases involving non-human hosts. WGS contain more mutations and offer greater phylogenetic resolution compared to single genes[7]. The development of novel models and methods that harness the plethora of genetic data, WGS and partial alike, could lead to stronger insights in pathogen control such as epidemic origins[8], geographic spread[9,10] and cryptic transmission[11,12].

Among these pathogens is rabies virus (RABV), one of the most well-documented zoonotic viruses, which is responsible for many local epidemics and an estimated 59,000 human deaths annually[13]. RABV is perhaps unique in how it has circulated in a large diversity of mammals over thousands of years, although 99% of human cases are caused by household dog transmission[14]. Despite much of Western Europe being rabies-free, 109 countries still battle the deadly disease[15,16]. There is a constant risk of RABV re-emergence by pet importation, travel to rabies-endemic areas, and wildlife transmission, which emphasizes the global need for a One Health, multidisciplinary and multi-national effort in rabies control[6,17]. Programming for epidemic control and prevention often relies on phylogenetic analysis for canine vaccination[18,19], which estimate RABV genealogies to infer spatio-temporal characteristics of rabies spread. This relies on global efforts in sample collection and sequencing[20].

Despite the global objective of zero human RABV cases by 2030[21], rabies remains a neglected disease primarily affecting low and middle-income countries, and there is suboptimal funding, disease investigation, and programming dedicated towards elimination[13]. This has stalled the reallocation of sequencing efforts towards WGS and leaves

[1]Institut Pasteur, Université Paris Cité, Lyssavirus Epidemiology and Neuropathology Unit, F-75015 Paris, France. [2]Department of Microbiology, Immunology and Transplantation, Rega Institute, KU Leuven, Leuven, Belgium. [3]World Health Organization Collaborating Center for Reference and Research on Rabies, Institut Pasteur, Paris, France. [4]Institut Pasteur, Université Paris Cité, Bioinformatics and Biostatistics Hub, F-75015 Paris, France. ✉ e-mail: andrew.holtz@pasteur.fr; anna.zhukova@pasteur.fr

phylogenetic and phylogeographic analyses still reliant on sub-genomic fragments. These analyses have been used to study the origin and spread of canine-maintained RABV in various regions, but often require reducing sample sizes for computational ease (choosing specific genes[22–24] or whole genomes[25,26]) which can introduce bias and reduce geographic and genetic diversity. This trend in partial sequence availability is not unique to RABV, as several other viral diseases also have a historical diversity of partial sequences, including West Nile virus, rotavirus species, and dengue virus.

Past and present phylogenetic analyses of RABV reveal a major divide in virus sequences associated with *Chiroptera* (bats) and *Carnivora* (canine-maintained and North Americans raccoons and skunks)[27], which infers a host switch between these two groups of animals. Within this division three phylogenetic groups of sequences have been identified: bat-maintained RABV and raccoon-skunk-maintained RABV and canine-maintained RABV within the *Carnivora*[27,28] (Supplementary Fig. 3). It is believed that canine-maintained RABV most likely rose in Europe and Asia during a period of dog domestication[27,28]. From this point, canine RABV was able to flourish across the globe[29], generating to two major phylogenetic groupings, the old world and cosmopolitan clades, that can themselves be subdivided into 44 subclades[26,29,30]. Despite the potential presence of bat-maintained RABV in the Americas pre-European colonization, canine-maintained RABV is believed to have emerged in Europe and then spread to the Americas following European colonization between 1642 and 1782[26–29].

This study proposes a novel gene concatenation method that takes advantage of the historical diversity of partial gene submissions to analyze the spread of canine-mediated RABV. We use 14,752 sequences to create a concatenated alignment, estimate a phylogenetic tree with 10,044 canine-mediated sequences and, from this, infer the ancestral dispersal around the world. Our findings are not only precise but, for the first time, provide ancestral geographic reconstruction on a global level across clades, regions, and countries. This further enables us to identify episodes of human-mediated introductions of RABV around the globe and inspect how European colonization starting in the fifteenth century impacted the spread of canine rabies. This not only provides valuable historical information for reconstructed transmission paths of RABV globally, but provides, for the first time, a useful phylogenetic framework for pathogens with heterogeneous sequencing data and to help inform introduction control policy.

## Results

### RABV sequence and metadata acquisition and composition
All 25,787 available sequences for the five genes of the RABV genome were downloaded from the NCBI Virus database[5]. After quality control, the RABV data set was reduced to 14,752 sequences that spanned 121 countries and were extracted from 192 different host species from 1972 to 2020. The multiple sequence alignment (MSA) comprised a concatenation of the five genes of the RABV genome (Fig. 1a and Supplementary Table 2). The maximum-likelihood (ML) tree from this MSA (Supplementary Fig. 3) revealed three major phylogenetic groups with bootstrap values greater than 0.95 corresponding to bat-, skunk-/raccoon-, and canine-related RABV, consistent with previous global RABV analyses[26,29]. The tree topology is spatially structured with clades, while gene fragments had a very low impact on sequence clustering (Supplementary Fig 3: color tips for gene fragments and simplified clade), indicating that sequences are not clustering by genetic region. All 10,209 sequences in the canine-related cluster were extracted and used for the remainder of the study.

*Canis familiaris* and genus *Vulpes* (i.e., foxes) were the most common sources of canine-clustering sequences, accounting for 51.9% and 8.4%, respectively. Other families, including the *Bovidae, Hominidae, Mustelidae, Felidae, Mephitidae*, and *Herpestidae*, contributed the remaining sequences. The location of the sampling impacted the

predominant source of sequences, with most European sequences coming from the genus *Vulpes* and those in Asia and Africa primarily coming from household dogs. In Asia, the *Hominidae* and *Mustelidae* families, particularly the Chinese ferret badger, were major sources of sequences (Fig. 1b, d). Since the 1970s nearly all continents have experienced an exponential increase in RABV sequence submissions to the NCBI Virus database. This increase is most notable for the N gene and the G gene, while WGS have experienced a slower rate in submissions, especially for Europe and Asia (Fig. 1d). WGS still only represents 13% of sequences and 62% of countries. This demonstrates the quantity of partial sequence diversity available and the potential loss of phylogenetic signal and resolution that occurs when only using WGS for phylogenetic studies. In addition, studies favor certain genes depending on which country/continent the study is located in. For example, proportionally, Europe sequences mostly the N gene, while Asia sequences many samples for the G, M, and P genes. As a result, only using the N gene or WGS introduces systematic sampling bias into any phylogeographic analysis by ignoring sequences that exist for other regions of the genome.

### Using all five gene fragments increases phylogenetic signal and tree-time calibration precision
Our ML tree of the 'canine' data set of 10,209 sequences containing all five genes confirms the clustering of canine rabies cases previously detected by Bayesian phylogenetic tree reconstruction using WGS (Fig. 3)[26]. The tree topology is spatially structured by countries of origin, while gene fragment and host species seem to have a very low impact on sequence clustering, providing evidence that rabies transmissions are more defined by geography than host species.

To better investigate the spatio-temporal spread and history of rabies across the world, we dated the phylogenetic tree, comparing evolutionary rates across the entire genome and for individual genes. The estimated evolutionary rate for whole-genome sequences was $2.00 \cdot 10^{-4}$ substitutions [95% CI: $1.95 \cdot 10^{-4}$; $2.22 \cdot 10^{-4}$] per site per year, while the rate for individual genes ranged from $1.9 \cdot 10^{-4}$ to $2.4 \cdot 10^{-4}$ (Supplementary Fig. 5), indicating little difference in evolutionary rate. This is consistent with previous estimates[26,28]. By applying the WGS evolutionary rate to the tree using LSD2 (where 163 sequences were removed by outlier removal), we dated the tMRCA of canine-related RABV to 1356 (95% CI: 1301; 1403). This estimate narrows previously published estimates in Troupin et al. (1308–1510) by 101 years[26] and in Velasco-Villa et al. (1273–1562) by 187 years[28] (Fig. 2). In general, our methods resulted in older age estimates for the tree compared to the findings of the other two studies.

The improvement in precision of dating is evident in previously established major clades[29,31] compared to Troupin et al.[26] and Velasco-Villa et al.[28] (Fig. 2). Our mean divergence time estimates were also consistently older, which can be explained by including more and older sequences ancestral to the clade roots compared to the other two studies. To name a few, we estimated the emergence of the Asian clade back to 1561 (95% CI: 1524–1594), the cosmopolitan clade back to 1656 (95% CI: 1627–1683), the Africa-2 back to 1799 (95% CI: 1761–1832), and Africa-3 back to 1723 (95% CI: 1697 and 1752). Notably, all point estimates of major clade emergence, with the exception of the cosmopolitan clade, fall within the 95% confidence intervals reported in Troupin et al.[26] (321 whole-genome sequences) and Velasco-Villa et al.[28] (266N gene sequences) as shown in Fig. 2 and Supplementary Fig. 5. The date difference for emergence of the cosmopolitan clade can be partially explained by the inclusion of more partial sequences in this clade clustering closer to the clade root.

### Subsampling method provides confidence in phylogenetic analysis and dating
To test the confidence in phylogenetic estimation, we performed country-aware subsampling on the original canine tree for five unique

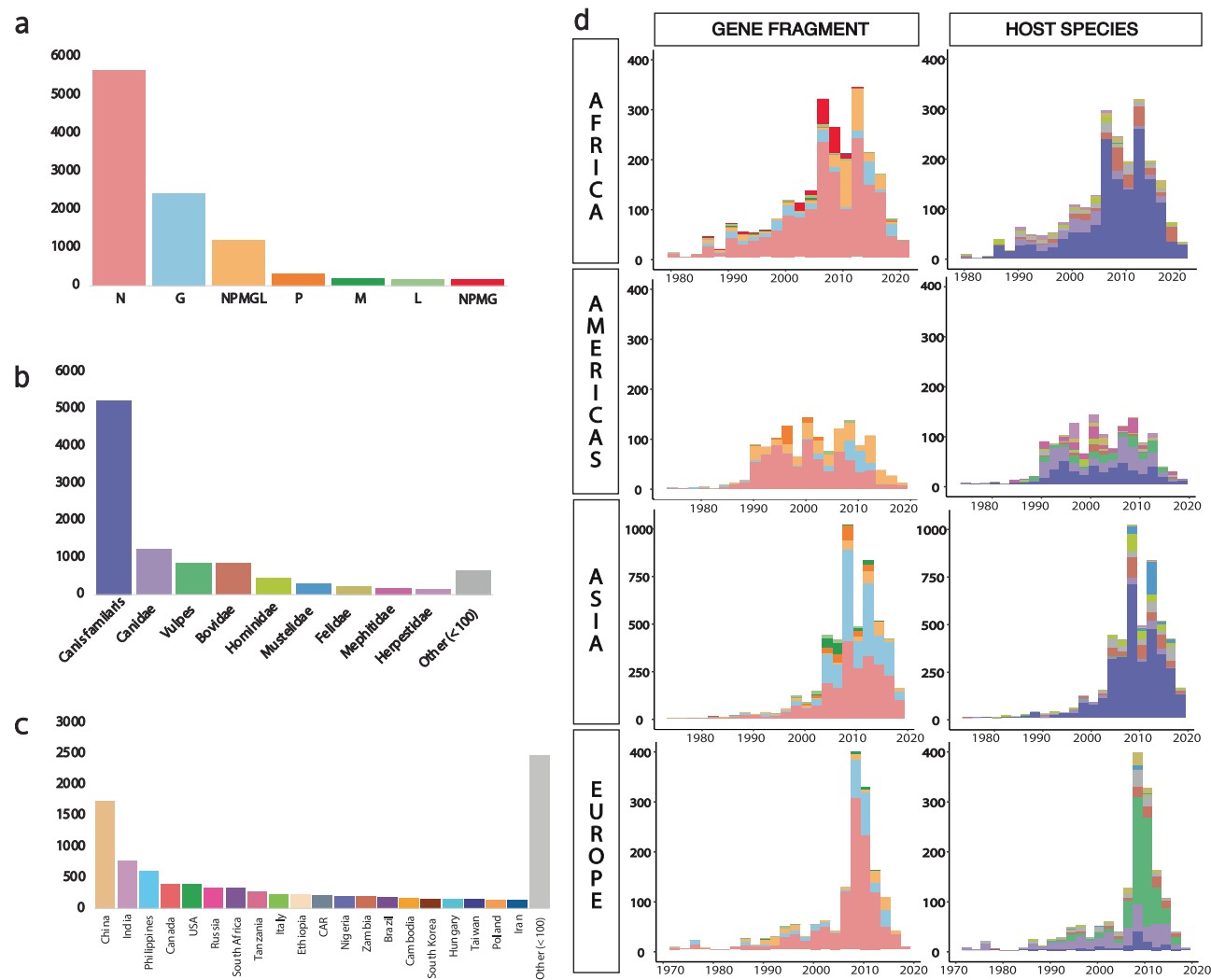

**Fig. 1 | Metadata exploration of the canine RABV sequences.** Composition by (**a**) gene fragment, (**b**) host species, (**c**) country of origin, (**d**) evolution of sequence deposition over time by fragment and host sorted by continent represented by stacked bar plots showing accumulated totals per year. Color definitions of host species and gene fragment are seen in plots **a** and **b**.

replicate sequence data sets of 5500 sequences each. To ensure the validity of phylogenetic analysis using the efficient but less thorough method[32] of FastTree for the full-canine tree, we performed a phylogenetic analysis using the more accurate IQ-TREE2 method (using an evolutionary model with gene-fragment partitioning) and compared the trees with the non-subsampled FastTree topology by triplet distance (where 0 corresponds to identical tree topologies and 1 to no triplet in common). Between the 5 subsamples and the full-tree, only 0.60–0.64% of tip triplets (Supplementary Table 3) varied between IQ-TREE2 with gene partitioning and the FastTree analysis (normalized triplet distance). This validates the fast method with concatenated gene sequences. Molecular dating of subsamples also further validates the tMRCA found for the larger tree. We found all tMRCAs for subsamples (ranging from 1368 to 1375) were very precise and all are located within the 95% CI of the full-canine tree tMRCA estimation, 1301–1403 (Supplementary Table 3, tMRCA).

**Purification and diversifying selection does not have a substantial impact on RABV tree dating**

To investigate the impact of selection pressure on dating, we used the aBSREL, FEL, and MEME methods in HyPhy and found evidence of purifying selection on 2793 sites out of 3490 variable sites tested on a WGS subsampled tree with 236 sequences from Troupin et al.[26]. We

also found evidence of diversifying selection on 24 sites ($p$ value < 0.01): (101, 436, 506, 587, 605, 633, 639, 716, 748, 839, 891, 969, 971, 1150, 1164, 1428, 1438, 1567, 1823, 1907, 3511, 3571, 3594, 3612). Notably, however, branch length-estimation for purifying and diversifying selection did not remarkably impact tree branch lengths and tMRCA estimates when compared to the original tree (intersecting confidence intervals) (Supplementary Fig. 4).

**700 Years of canine-maintained RABV spread**

To investigate the geographic transmission of canine RABV, we inferred the ancestral locations at the country level on the full and subsampled trees with PastML (https://github.com/amholtz/GlobalRabies/tree/main/data/ACR_Results)[33]. We created a consensus tree with 6,096 sequences by taking the union of all sequences in subsampled trees and reducing the full-canine tree through pruning. The clade ancestral estimates that were consistent between the aggregated subsamples and the full tree are shown in Fig. 3. For the first time, we estimated the country origin of 34 out of the 44 identified canine-mediated clades in the consensus tree, with ancestral estimates as well for their internal nodes, revealing historical transmission between and across the clades (Supplementary Table 4 and Fig. 3).

To resolve location uncertainty that remained from the country level ancestral character reconstruction (ACR) we reconstructed

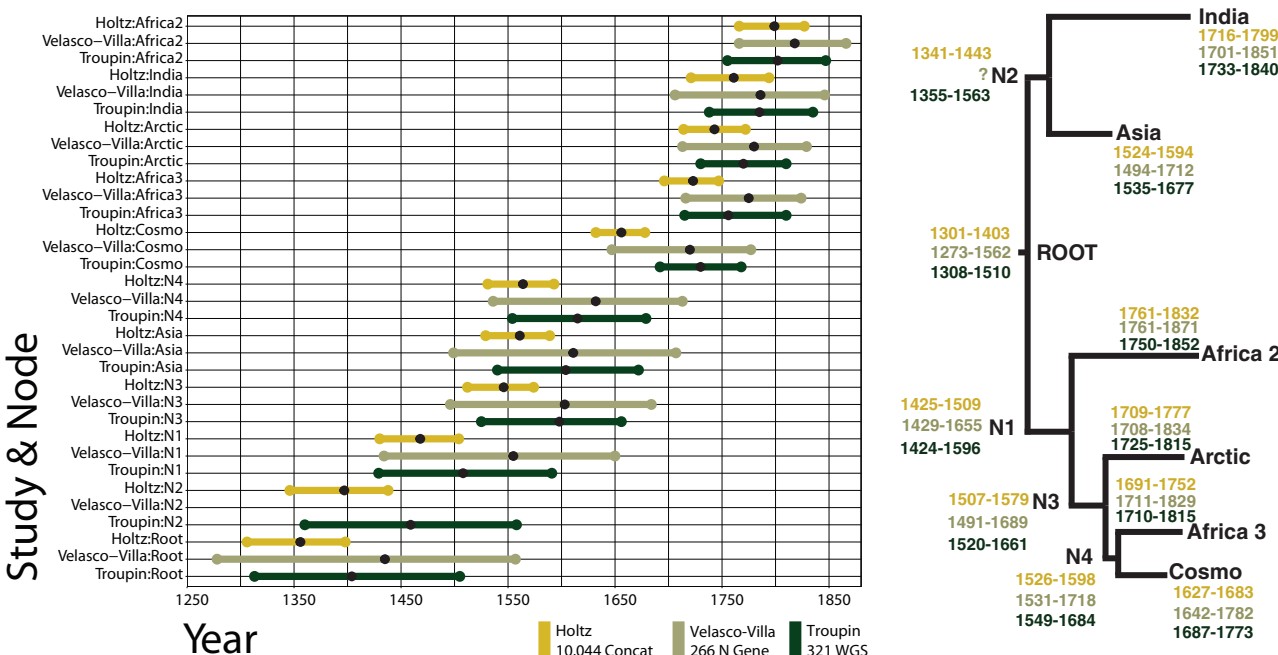

**Fig. 2 | Comparison of previously published global time-calibrated trees of canine-mediated RABV.** The present study is shown in comparison with two previously BEAST reconstructed timetrees, Velasco-Villa et al.[28] used 266 nucleo-protein (N) gene sequences and Troupin et al.[26] used 321 WGS. 95% Confidence intervals of all node defining-major clades and ancestral nodes to the root are plotted. The bullseye on each line represents the reported point estimates. Locations of each node (presented by N) on the tree are shown on the right, as well as the numerical confidence intervals for the reported date estimate. Velasco-Villa et al.[28] does not report the date estimation or confidence interval for N2.

ancestral states again, by grouping countries into 23 geographic regions (18 of which are represented in our data set) previously defined by the World Bank grouping[34,35]. Out of 3114 internal nodes, the PastML MPPA method estimated a unique region for 3091 (99.26%) of the internal nodes. By grouping, we were able to estimate regional origins for nodes that are unresolved in the country reconstruction visualization (Fig. 4a). We estimated the regional origin for 100% of the previously defined canine-mediated clades[26] and the regional origin for 38/44 (86.7%) of their parental nodes (Supplementary Table 4).

Consistent with previous studies, we estimate the presence of canine rabies in Eastern Asia by 1561 (1524–1594, light brown node "1561 Eastern Asia" in Fig. 4a) and in Southern Asia by 1760 (1716–1799, light blue node "1760 Southern Asia", Fig. 4a). From Eastern Asia, we estimate that there have been multiple introductions to South-East Asia (SEA, pink nodes in Fig. 4a), revealing a continuous reintroduction pattern in the region (Fig. 4b). Independent from the SEA clusters are cases of RABV from China and South Korea (Arctic-AL2, light brown node "1866 Eastern Asia" in Fig. 4a) that cluster with arctic sequences. Its ancestral origin was estimated to be China, dating back to 1866 (1843–1885) (Supplementary Table 4 and Fig. 3).

By regional grouping, we estimated the emergence and maintenance of RABV in West Africa by 1799 (1761–1832; 95% CI, salad-green node "1799 West Africa" in Fig. 4a; Supplementary Table 4 and Fig. 3). We can further define this emergence on the country level, where we see three diverging events within this West African clade associated with sequences from Central African Republic, Senegal, and Nigeria. The major node estimated to be Nigeria emerging in 1933 (1924–1943) contains 97% of the Africa-2 sequences.

We estimated the date of emergence and maintenance of the cosmopolitan clade slightly older than previous studies, 1656 (95% CI: 1627–1683) compared to 1720 (95% CI: 1642–1782)[28] and 1730 (95% CI:1687–1773)[26] (Fig. 2). This is the first study that estimates the ancestral geographic origin of the downstream subclades. At the regional level, our method estimates the ancestral origin as Northern America, dating to 1656 (Fig. 4, yellow node "1656 Northern America").

Northern America most likely represents sequences from early European colonization of the Americas. For subsequent subclades, we are able to confidently infer the country origin of 23 out of 25, and the regional origin of 25 out of 25 of the previously defined cosmopolitan subclades (Supplementary Table 4).

We can further visualize the subsequent emergence from Northern America (USA) to Central America in 1851 (1836–1864, pale green node "1851 Central America" in Fig. 4a), leading to cases in Cuba in 1905 (1898–1911, maroon node "1905 Caribbean", see also Figs. 4b and 3). We also infer that the AM2a subclade emerged via dissemination from the USA to Mexico in 1851 (1836–1864). Additional introductions in Southern America can be seen throughout the cosmopolitan clade during different periods but do not seem to be widespread or well sampled (Supplementary Table 4, Figs. 3 and 4).

The transmission of cases of canine rabies in Eastern and Southern Africa can be traced back to an introduction from Northern America (early European colonies) (Fig. 4) to an intermediary position (unresolved between Western Asia or Eastern Africa) in 1805 (1791–1818), and eventually to Eastern Africa in 1826 (1812–1837). We can see repetitive transmissions between Namibia, South Africa, and Zambia, representing Southern and Eastern Africa, supporting previous studies[36].

From the same intermediary region in 1805 (1898–1911), canine rabies further spread to Western Asia in 1809 (1794–1823), and eventually to Europe leading to the emergence of European fox rabies in 1873 (1863–1881, green node "1873 Eastern Europe" in Fig. 4). This is a major node of interest since it is a reflection point that led to non-imported European cases of Rabies. The CA1 subclade which contains sequences from Russia, Ukraine, Latvia, Tajikistan and even shows transmission to parts of China and Mongolia[37,38] is estimated to have emerged in 1924 (1913–1935) via Russia (Fig. 3).

Canine rabies present in the Western Europe and Central Europe subclades most likely originated from Germany in 1948 (1940–1954) and from either Germany or Poland in 1968 (1962–1973). Interestingly, these two subclades share an ancestral node dating back to 1931

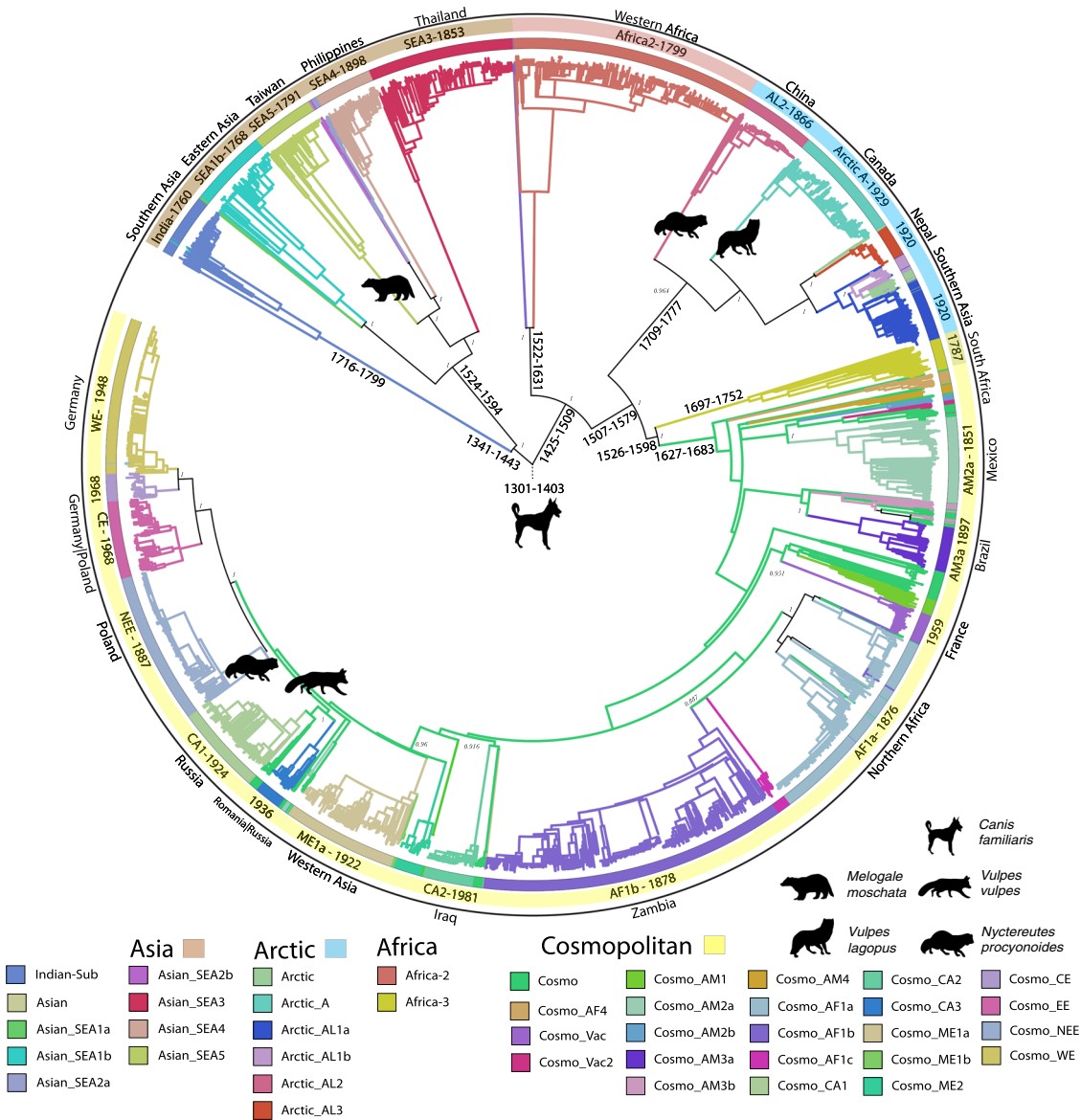

**Fig. 3 | ML consensus phylogenetic tree of canine RABV sequences (*n* = 6096).** Confidence intervals of LSD2 dating are provided on the most internal nodes. Branch colors and colorstrip present estimated ancestral clade by ancestral character reconstruction (ACR). Unresolved branches are black (the legend is shown below the tree). Names of clades and date of clade tMRCA are provided if space is sufficient. The outer label presents the country or regional ancestral estimate for that clade. *Canis familiaris* is the most dominant host species in the tree. Clades with a majority other than *Canis familiaris* are shown on the clade-defining branch. Bootstrap support from 1000 replicates is shown on informative clade-defining nodes (silhouette images of animals are sourced from http://phylopic.org/. These are available for reuse under the CC0 1.0 Universal Public Domain Dedication license).

(1921–1938) estimated as either Germany or Poland. Poland is also the inferred ancestral origin of the NEE subclade in 1887 (1873–1899) leading to cases in Lithuania, Latvia, Estonia, and Poland (Supplementary Table 4 and Fig. 3). Additional patterns of rabies emergence can be seen on the PastML compressed visualization (https://github.com/amholtz/GlobalRabies/tree/main/data/ACR_Results)[33] and the table of significant clades (Supplementary Table 4).

**European colonization likely contributed to RABV transmission**
For the first time, we reveal how historical colonization could have shaped canine rabies spread around the world by reconstructing ancestral scenarios on the colonization history of countries of isolation. Major nodes closest to the root are unresolved; however, the colony with the highest probability (35.9–38.5%) for these unresolved nodes is the British Empire (Fig. 5). This indicates that colonization could have played an instrumental role in the expansion of this strain

of RABV. We also find that the French Empire is the inferred ancestral location (marginal probability of 84.1%) for the Africa-2 clade by 1799 (Fig. 5, "**"). and the inferred ancestral location (marginal probability of 99.8%) for the majority of the SEA3 clade by 1942 (Fig. 5, "*"). Further, we estimate the Spanish Empire as the ancestral location (marginal probability of 85.5%) for the cosmopolitan clade (Fig. 5, "***") and subsequent AM2a subclade in Latin America. It is important to note that Mexico is grouped as the Spanish Empire in this analysis and is grouped as Northern America in the regional analysis. Although this shows how colonization could have sustained RABV spread in the world, there are many variables and factors not accounted for, such as early global trading companies[27,28,39].

**Identification of human-mediated transmissions of canine RABV**
To identify human-mediated events of rabies transmission, we consolidated transmissions in the full-canine tree across long distances

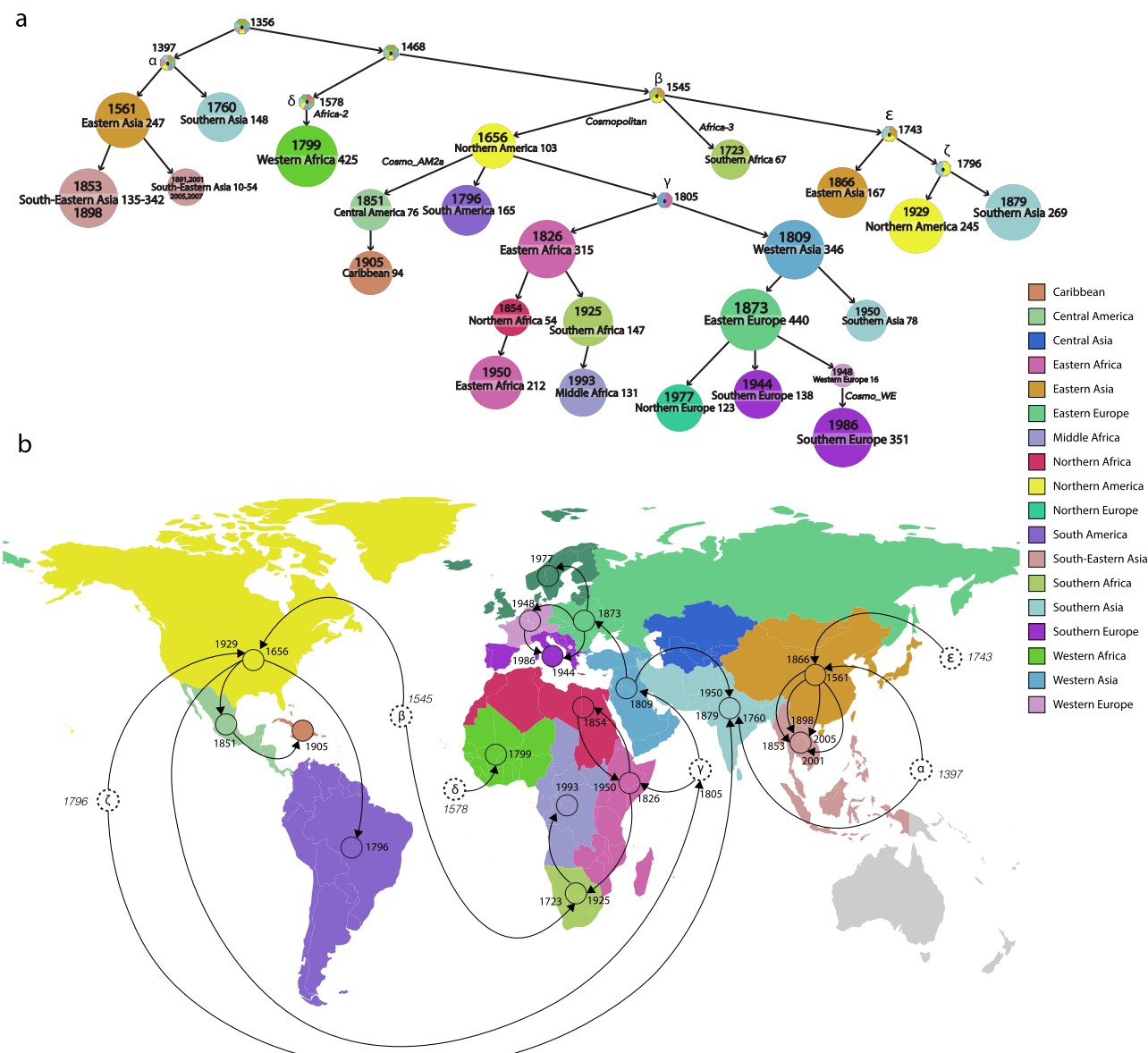

**Fig. 4 | Regional ancestral character reconstruction for canine RABV tree from PastML MPPA. a** Compressed visualization of consensus tree (of 6096 samples), where the parts of the tree without regional changes are clustered. For each cluster, the date of the most ancestral node, its region and the number of samples represented are shown. The clusters are colored by region (the color legend is shown on the right). **b** RABV regional spread represented on the world map. Dates represent the end of the branch, while italicized dates represent the dates of unresolved nodes (dotted circles with greek letters corresponding to **a**). Dotted circles without a prior path are unresolved up until the root, which is dated at 1356. Confidence intervals can be seen in Supplementary Table 4. Circle locations represent the region rather than country of origin. For example, 138 of the sequences in South-Eastern Asia are isolated from the Philippines, even though the circle is located in Thailand. Grayed regions represent regions with no samples in our data set.

(greater than 2000 km) and rate of spread (faster than 100 km/year) and/or over large bodies of water. Of 14,640 parent-to-child node transmissions observed in the tree, 1131 transmissions were identified between different countries, and 232 of these were to non-neighboring countries. Of these 232, we identified 43 transmissions of interest according to their speed of transmission (distance over time) and whether the transmission crossed a water barrier (Fig. 6 and Supplementary Table 5). The majority of the transmissions identified have not been previously reported and indicate cryptic transmission by human mobility. It is important to note that nodes without ancestral country estimates are ignored in this analysis.

## Discussion

We present a phylogenetic pipeline that harnesses the information from partial and whole-genome sequences and their metadata to investigate the spatio-temporal dispersal of historic epidemics and the role of humans in their spread. Using this analysis, we were able to reveal the patterns and timing of the global spread of RABV.

The challenge in analyzing sequence data from pathogens that have existed for a long time is the variability in the quality and completeness of the data, such as the presence of partial genes versus complete genomes, the accuracy of the sampling date and location information, and the unequal representation of different geographic areas. When performing phylogenetic analyses on such data, a choice needs to be made between discarding some data by selecting a more homogeneous and higher-quality subsample (e.g., only the whole genomes or a particular gene) to avoid potential noise and keeping more or even all the data to increase the inference power. The former approach allows using more complex and hence time-consuming methods, while the latter requires faster methods as a result of many

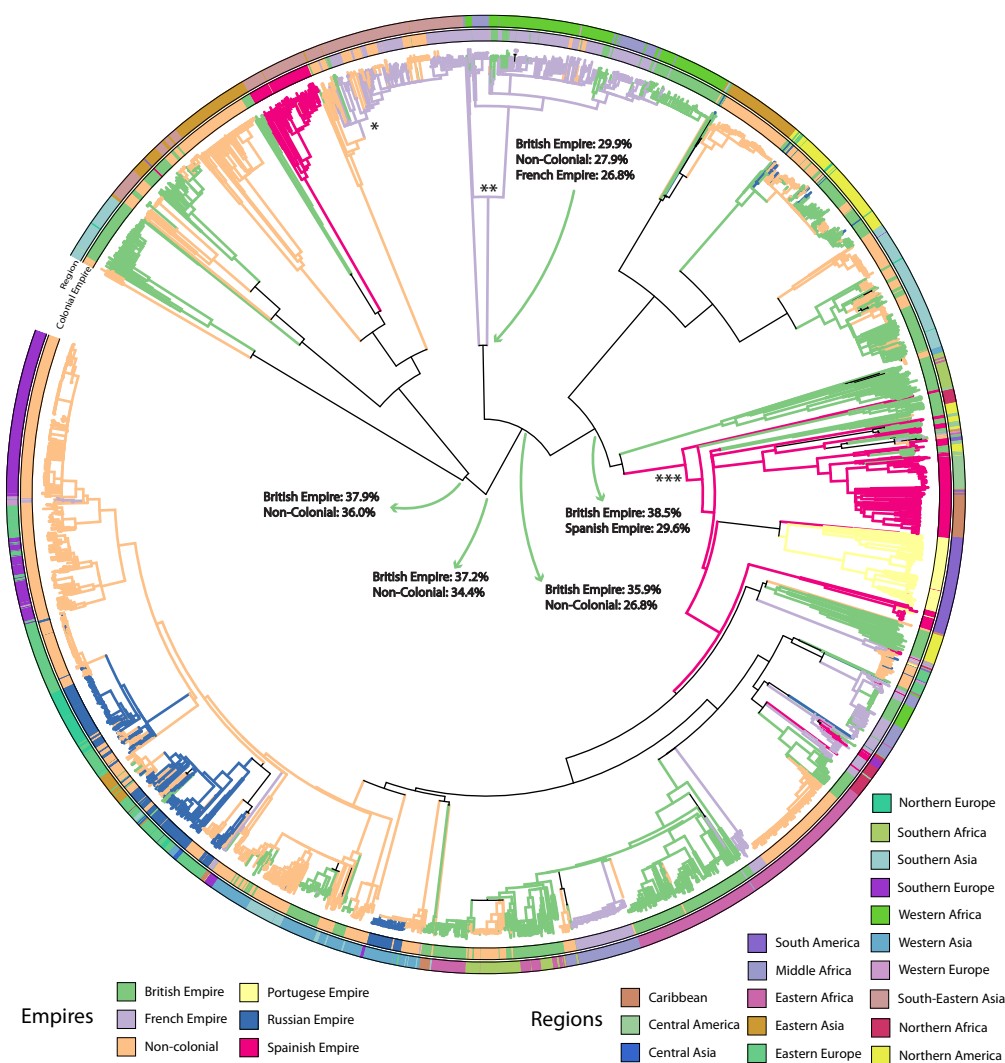

**Fig. 5 | RABV transmission history organized by colonial powers from 1600 to 1950s.** The consensus tree represents 6,096 sequences, colored according to the inferred colonial empires at the nodes (the color code is explained in the legend on the bottom). The color strips around the tree represent historical empires (inner) and regional grouping (outer). Nodes of interest indicated: (*) 99.8% French Empire- 1942, (**) 84.1% French Empire-1799 (***) 85.5% Spanish Empire 1656. The visualization is performed with iTOL[73] on a PastML-annotated tree.

more sequences being available for inclusion. Using a larger data collection for phylogenetic and phylogeographic inference will result in the consideration of a wider range of sampling times, as well as data from additional countries or regions, allowing for a more detailed analysis and conclusion. Previous RABV studies[26,29,39,40] have opted for the former approach, focusing on smaller (hundreds of sequences) data sets representing specific partial genes or WGS, which were analyzed with Bayesian or ML inference methods. We investigate the opposite approach and shows its advantage in terms of inference power.

Our analysis is based on more than 10,000 RABV partial and whole-genome sequences available in the NCBI Virus database[5]. Our pipeline employs a concatenated alignment of the five RABV gene fragments, with gaps for the regions absent in partial sequences (Supplementary Fig. 2). The phylogenetic analyses were performed with time-efficient inference methods suited for large data sets: an approximate maximum-likelihood phylogenetic analysis using FastTree, time-scaling with the least-square dating method of LSD2, and ancestral geographic character inference using a maximum-likelihood method implemented in PastML. We validated our pipeline by comparing its results to those from previous studies, to phylogenetic analyses using subsampled data sets and to those utilizing more complex evolutionary models accounting for potential evolutionary rate changes between genes and for potential selection pressure:[26,29,39,40] Using a larger data set, we obtained a compatible tree topology and compatible (between different trees and evolutionary models) but more precise dating (Fig. 2).

Additional challenges in pathogen spread analyses are posed by sampling bias in dating and country representation, which can influence phylogeographic reconstructions. The fact that with long-lasting epidemics (e.g. for centuries) even country definitions might change over time (e.g. from the British Empire to the United Kingdom; UK) further complicates the story. Our method attempts to remove sampling bias as a possible factor in our ancestral character reconstruction in three ways. First, we established a subsampling protocol which removes sequences from oversampled countries, since it is well known that ACR is heavily influenced by the number of sequences per character state[41]. In addition, we report ACR on both regional and country levels since there are some regions with less representation by countries. Finally, we were able to include subgenomic fragments independent of which gene is sequenced by representing the five genes of the RABV genome.

Employing time-efficient methods allowed us to harness the plethora of information of a very large data set to achieve a precise

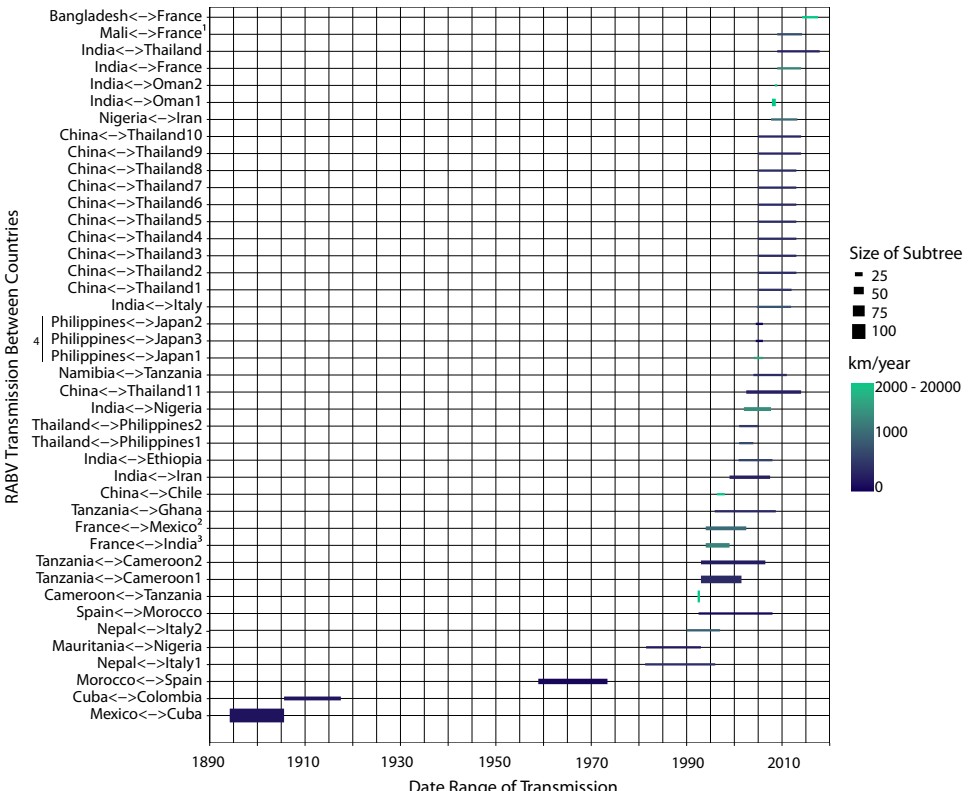

**Fig. 6 | Inferred human-mediated introductions identified on the phylogenetic tree by date.** Human-mediated transmissions are presented by time of transmission between estimated parent and child dates. Transmissions between two countries are shown on the y axis. In some cases, there are multiple transmissions between two countries. The size of the subtree (descendants) from the transmission is shown by line width. For example, the child node for Mexico-Cuba has 109 descendants. Transmission speed is represented by the geographic distance (km) over one year and is calculated by the distance between the most-populated city between parent and child and the branch length. Human-mediated introductions are defined as transmissions with speed faster than 200 km/year and between two countries that are more than 2000 km apart or separated by a body of water (e.g. seas/oceans). Certain transmissions (identified by superscripts) can be linked to documented introductions (see Discussion). This figure depicts importations between two countries rather than indicating the direction of importation due to limited sequence availability.

estimation of tMRCA of modern canine RABV and of dates of emergence for each of its 45 clades (Supplementary Table 4 and Fig. 3). Our estimates are compatible with previous studies by Troupin et al. (whole-genome data)[26] and Velasco-Villa et al.[28] (N gene) (Supplementary Table 4) (Fig. 2). However, our confidence intervals are almost two times narrower. For instance, we found the tMRCA to be 1356 (95% CI: 1301–1403), while Troupin et al. found the tMRCA to be 1404 (95% CI:1308–1510)[26] and Velasco-Villa et al. estimated 1435 (95% CI:1273–1562)[28] (Fig. 2). In addition, we also validated that, although present, purification selection[42,43] does not impact time-tree calibration of RABV. Interestingly, there are historical records of rabies predating our date of emergence in ancient Rome, Greece, and Egypt[44–46]. We believe that over history, there have been many different clades of RABV in circulation, and these versions were outcompeted by more infectious renditions. Our analysis is relevant for the current RABV that is in circulation today, which spread and replaced all other circulating RABV strains, aided by human urbanization, global trade, and dog introduction to the human home[29].

Despite the long history of canine-mediated RABV, it does not seem to be strongly impacted by recombination. This is due to the fast fatality rate and infection/transmission dynamics, which make coinfection of different virus types rare[47]. In contrast, bat communities can experience non-fatal RABV infections, allowing for multiple strain coinfections. Our study also shows a distinct geographic distribution of canine RABV genetic variants with minimal local variation. Therefore, any recombination would likely occur among closely related viruses and remain concealed in large-scale phylogenetic analyses, like those in our analysis.

Considering these factors, we performed for the first time geographic ancestral character reconstruction on internal nodes of the global canine RABV phylogenetic tree. These estimates, however, are reliant on sequence availability and must be represented under this light. For example, we do not have records for historical rabies sequences that could have been circulating around the time of canine rabies expansion. Previous colonizing powers, like the UK, eradicated rabies before RABV sampling was possible[45]. The only representation we have of UK sequences are those left behind in their early colonies, such as the USA and Canada. In light of our analysis, we believe there is a cryptic node before the spread of canine rabies to Northern America in 1627–1683, which represents main European colonizers or global trade organizations. The cosmopolitan clade would originate from this node, which represents old, pre-eradication, European sequences. The node that defines Northern America would only lead to subsequent introductions to Central and South America.

The role of historical events in shaping RABV dispersal patterns is a central question that remains largely unanswered by measurable, quantitative approaches. To test the role of European colonization on historical RABV dispersal, we reconstructed the ancestral dispersal of canine RABV by the British, French, Spanish, Portuguese, and Russian colonial powers (Fig. 5). We observe some clear trends: the French Empire is associated with the spread of canine RABV in West Africa, the Spanish Empire with the spread in Central and South America, and the Russian Empire with the spread in Eastern Europe and Central Asia. The impact of the British Empire is, however, the most dominant throughout the phylogeny (Fig. 5) and is the most likely location for most ancestral nodes. This suggests that the British Empire played the

most important role in the spread of canine RABV across the world. It is also the most likely location for the parental node shared by the Cosmopolitan and Arctic clades (Fig. 5) as well as defining the parental node that defines Africa-2. Further investigation could include additional international players of the time, such as trade networks by the Dutch East India Company (DEIC; 16th and 17th centuries) and the Chinese Ming Dynasty (fifteenth century)[26,28]. Indeed, the DEIC could be associated with spread of the Cosmopolitan clade, since the parental node (1526–1598) to the North American introduction has two branches: (1) hypothesized North American introduction 1627–1683, and (2) hypothesized introduction in South Africa in 1697–1752. The DEIC established trading routes with the USA in 1607 and with South Africa in 1652. Dogs were gifted for purposes of hunting, dog fighting, and companionship, and could have exacerbated the transmission of rabies around the world[48]. Similarly, the Zheng fleet of the Ming Dynasty (fifteenth century) could help explain introductions of RABV from an early unresolved ancestral node between 1341–1443 to Eastern (1524–1594), South-Eastern (1553–1623), and Southern (1716–1799) Asian nodes (Fig. 4a and Supplementary Table 4). The Zheng fleet established trade routes, often transporting animals, in Southern and Eastern Asia and the Arabian Sea during the 1400 s potentially introducing RABV to Eastern, Southern, and South-Eastern Asia[49]. However, we did not find evidence of early rabies transmissions from Eastern Asia to the Arabian Peninsula and Eastern Africa (Fig. 4a) despite the establishment of Zheng trade routes between 1405 and 1433.

The human-mediated importation of RABV was not unique to colonial times. Each year, there are instances of human-mediated introduction of RABV via immigration, travel, or dog importation[17]. We detected human-mediated transmissions on a time-calibrated tree with country annotations by searching for transmissions over branches representing long geographic distances in a short time frame[50]. Detectable instances must have a sequenced case in both the arrival (descendent) and departure (ancestor) country or some additional information on travel history[51]. Due to a lack of sequencing, the direction of importation cannot be reliably inferred. For the importation between France and Mexico (Fig. 6, superscript 2), we identified a known case of RABV which was sequenced in France on a sample obtained from a traveler who had returned from a trip to Mexico[52]. Therefore, the sequence labeled as 'French' actually represents the imported case from Mexico rather than a lineage currently circulating in France. This pattern could be present for other samples as well, such as cases between Spain and Morocco. Future work could incorporate travel history into the analysis to inform patterns of transmission[51]. Additional importations in our analysis (superscripts 1, 3, 4) correspond well to documented cases from historical records[52]. This approach is, therefore, not comprehensive but does provide a time and geographic range for possible human-mediated transmissions. In addition, using distance between most-populated cities in each country is an oversimplification of travel between two countries. Certain transmissions could be identified as significant if two cases might be from bordering areas (for example, a transmission between Tehran and Moscow might appear as significant (>3000 km) even if both cases were collected next to the Azerbaijani border, in Russia and Iran respectively). We do observe a trend that long-distance transmissions have increased over time. It is tempting to suggest, therefore, that long-distance canine RABV transmissions increase with globalization, although this is highly biased by the fact that we have greater resolution closest to the tips of the tree as a result of more sequencing.

Although we aimed to reduce sampling bias by subsampling our data set by country, utilizing all genes in NCBI, and grouping by region to reduce focus, our study still contains biases. A recent study sheds light on the bias in lineage definitions for RABV and introduces a proposal for new lineage definitions using dynamic nomenclature. Furthermore, a recent review, addresses the geographic bias resulting from the under-representation of sequences in countries with high

numbers RABV-related deaths, such as India, Afghanistan, Russia, and middle/west Africa[1]. This bias requires the use of statistical methods to account for underrepresented countries. If all countries with RABV cases submitted sequences uniformly, the need to subsample would be less, and overall, the confidence of RABV phylogeographic studies would be stronger. In addition, it is important to note that while the number of RABV sequences shared on NCBI has increased since 1970, there has been a significant decrease in submissions in recent years. There are many factors that could explain this decrease including the COVID-19 pandemic, political issues within countries or decreased global funding for neglected tropical diseases. To address this issue, continued global and One-Health efforts to increase RABV surveillance and sequencing are necessary[6], as is the open sharing of that data to enable effective genomic surveillance of RABV.

By concatenating the five genes of the RABV genome and using time-efficient methods, we achieve the most precise date and geographic estimates for the emergence of canine-maintained RABV yet reported. This is the first investigation of this type on the global spread of canine RABV and provides valuable resources for epidemiological investigation of RABV introduction events and epidemic control. This concatenation method and dispersal history reconstruction will not only allow for a more precise understanding of global trends for rabies but also can be applied to other pathogens with a large deposit of partial sequences for phylogeographic investigation and dating purposes.

## Methods
### Data set compilation
A total of 25,787 sequences of rabies virus (RABV) were obtained from the NCBI Virus database[5] in September 2021. A quality check was conducted to remove sequences that were (1) missing date and country information, (2) older than 1972 (for sequencing quality), (3) identified as vaccine or laboratory strains, (4) or shorter than 200 nucleotides in length. As a result, 14,752 sequences were retained for this study. Sequence filtering is demonstrated in Supplementary Fig. 1, and a complete list of sequences with criteria for exclusion can be found in Supplementary Table 1 ('exclusion' column).

### Sequence alignment
An initial global alignment using a custom script (https://github.com/amholtz/GlobalRabies/blob/main/R/clean_RABV.R)[33] was used to define the sequences based on which gene their nucleotides correspond to. Sequences with more than 200 nucleotides in a gene-coding region, were sorted into gene-specific fasta files. For example, WGS were included in the N, P, M, G, and L files. All sequences in these independent files were then cut at the start codons (according to the reference genome) of the corresponding gene (top, Supplementary Fig. 2).

Each newly cut gene-specific fasta file (void of non-coding regions) was then aligned against the (also cut) corresponding reference sequence, using MAFFT (v7.505) with the FFT-NS-1 strategy, addfragments, and keep-length method[53]. This resulted in five multiple sequence alignments for each gene-coding region: N gene (1353 nt), P gene (894 nt), M gene (609 nt), G gene (1575 nt), and L gene (6429 nt). Once all genes were realigned, they were concatenated by ID using Goalign (v0.3.5)[54] column-wise in original order, introducing gaps (-) where gene fragments were missing (non-sequenced locations for each ID). The final product is a global multiple sequence alignment, called concat_seq_genes.fasta with a total length of 10,860 nucleotides (bottom right, Supplementary Fig. 2).

### Phylogenetic estimation, dating, and subsampling
**All RABV sequences (bats, skunks, canines).** FastTree (v2.1.11)[55] was used for the initial phylogenetic analysis of the 14,752 sequences to confirm the separation of the previously identified clades by

Troupin et al.[26]: (1) bats, (2) skunks and raccoons, and (3) canines and their myriad subclades (ran on 32 threads for approximately 1 h:15 min). To this end, the GTR nucleotide substitution model[56] and a discretized gamma distribution to accommodate among-site rate heterogeneity were employed, and bootstrap support values were estimated using default settings (Shimodaira-Hasegawa test[57,58]) (Supplementary Fig. 3).

**Canine sequences.** The canine-mediated RABV clade was identified and a subset alignment from the original was created ($n = 10209$). A canine-mediated tree was reconstructed from this alignment using FastTree (v2.1.11)[55] (ran on 32 threads for approximately 47 min). The evolutionary rate was estimated using whole-genome sequences exclusively to mitigate potential sequencing errors from gene-specific projects. To estimate the evolutionary rate, the least-squares dating (LSD2) method version 1.8.8[59] was used on the whole-genome-only canine tree without replacing the root and with default outlier removal. The resulting evolutionary rate was then provided as a fixed parameter in LSD2[59] to yield a time-calibrated tree for the full phylo-genetic tree with 10,209 sequences with outlier removal and with confidence intervals based on 1000 replicates. A total of 165 sequences were detected as outliers and removed, according to a Z-score of 3 (listed in Supplementary Table 1), resulting in a full-dated canine-derived phylogenetic tree with 10,044 sequences, referred to as the full-canine tree henceforth.

To visualize differences in rates of evolution according to each gene, phylogenetic trees using FastTree (v2.1.11)[55] were estimated using the previously reported settings, according to five gene-specific multiple sequence alignments. Gene-specific evolutionary rates were then determined using LSD2[59] following outlier removal (Supplementary Fig. 5).

**Subsampling.** The full-canine tree was subsampled to reduce geo-graphic bias towards countries with a disproportionally high number of sequences by employing a custom script (https://github.com/amholtz/GlobalRabies/blob/main/python/py_subsampling.py)[33] which removes sequences from oversampled countries by phylogenetic diversity[60] given an input tree. At each step of the algorithm, the tree tip with the shortest branch among those sequences corresponding to over-represented countries is removed. The process is repeated until a desired target number of sequences is reached for all the countries. As several candidates for removal might exist at each step (i.e. tips with the same, shortest, branch length), the tip to be removed is chosen randomly in such cases. The subsampling was repeated five times, generating slightly different (due to stochasticity) trees with 5500 sequences each. Phylogenies from these subsamples were esti-mated using IQTREE-2 (v2.2.2.2)[61] employing the GTR + I + G4 model (determined by ModelFinder[62]) with gene partitioning to account for potential variation in substitution rates by gene and evaluated with 1000 ultrafast bootstraps[61,63,64] (ran on 8 threads for 164 h:28 min). The resulting tree topologies were compared with the full-canine tree by measuring triplet distance[65,66]. The subsampled phylogenies were time-calibrated in the same way as the full-canine phylogeny and the tMRCA estimates were compared. Finally, the ancestral country reconstructions on the subsampled and full timetrees were compared (see below).

**Purifying selection model**
To quantify the effect of purifying and diversifying selection on the time calibration, we used the software package HyPhy[67] on the smaller data set of 236 whole-genome sequence subset from Troupin et al.[26] to lighten the computational load[26]. This subset is made of only WGS and its dating has been previously analyzed, such that it is useful to assess model variations. Site-specific rates of the numbers of nonsynon-ymous and synonymous (dN and dS) substitutions per set were

inferred in each codon using the Mixed Effects Model of Evolution (MEME)[68], Fixed Effects Likelihood (FEL)[69] and the adaptive Branch-Site Random Effects model (aBSREL)[70] in HyPhy. The dN/dS ratio was then used by each model to optimize and re-estimate branch lengths of the tree with a fixed topology (Supplementary Methods 1). The branch lengths of the initial tree topology were re-estimated under each model, and the resulting trees were then dated using LSD2. The root dates were compared to those of the initial tree (estimated under the GTR model) (Supplementary Fig. 4).

**Phylogeographic reconstruction and subsampling consensus**
Ancestral geographic characters of canine RABV transmission were investigated using the fast likelihood method available in PastML version 1.9.34[71] to reconstruct the historical spread of canine RABV. Marginal probabilities of locations of ancestral nodes were estimated given the geographic location of sampled sequences in a rooted time-calibrated tree. To assess the confidence in state estimates of internal nodes with respect to potential geographic sampling bias, we repeated PastML on each of the five independent and uniquely subsampled reconstructed and time-calibrated trees and compared the ancestral state estimates. Only ancestral state estimates with marginal posterior probabilities of 50% or greater were considered. A custom script (https://github.com/amholtz/GlobalRabies/blob/main/R/ACR_Sub_comparison.R)[33] was used to identify and compare nodes with identical cluster compositions between the subsampled and full-canine trees to create an aggregated consensus country-annotation (Supplementary Fig. 6 and Supplementary Methods 2). The consensus tree was created by pruning the full tree to keep the union of all sequences in sub-sampled trees, yielding a final tree with 6096 sequences (Fig. 3), and the aggregated ACR result was used as a new annotation for the con-sensus tree using PastML.

**Regional and colonization history ancestral character reconstruction**
ACR using PastML version 1.9.34[71] was repeated on the 6096-sequence aggregated tree for two geographic characters. The first defined the character using a World Bank regional grouping variable using R package countrycode[34] (Fig. 4) which reflects cultural and geographic similarities. This resulted in the grouping of 120 countries in the ori-ginal data set into 18 regions. The second defined the character according to the colonial history of the sequence country of isolation (Fig. 5). Colonial powers were considered from 1600 to 1950, resulting in five states: British, Spanish, French, Portuguese, and Russian Empires (not including USSR satellite states). Countries with more than one colonial history were considered a member of the empire that maintained the longest and largest control (for example, India is grouped by the British Empire, even though a small part was also a French colony for several hundred years). Countries that have never been colonized were grouped as 'non-colonial' (Supplementary Table 1). The aggregated tree was used since it has been subsampled to limit bias from geographic origin. For both character analyses, the marginal posterior probabilities approximation (MPPA) method[71], was used for ancestral character estimation. MPPA uses decision-theory concepts to estimate a unique state in tree regions with low uncer-tainty, and several states in uncertain ones (typically around the root).

**Introduction event exploration**
To visualize human-mediated transmissions of RABV, long geographic distances over short time intervals were located on the phylogeny using a custom script (https://github.com/amholtz/GlobalRabies/blob/main/R/introduction_events_FullTree.R)[33]. The package cepiigeodist[72] was used to calculate geographic distances between each most-populated city in the country. Character states from the phylogeographic analysis were considered only if there was a consensus between the aggregated subsamples and the full-canine tree. A branch of length Y with a

resolved consistent parent node state A and a resolved consistent child node state B was considered as a transmission from country A to country B over Y years. Transmissions within countries and between neighboring countries were discarded, resulting in 232 long-range transmissions. Of these transmissions, those over a distance of more than 2000 km or that were across a water barrier and had a transmission dispersal interval faster than 200 km/year were conserved (Supplementary Table 3), resulting in 43 transmissions.

## Reporting summary

Further information on research design is available in the Nature Portfolio Reporting Summary linked to this article.

## Data availability

Data were downloaded via NCBI Virus Database: https://www.ncbi.nlm.nih.gov/labs/virus/vssi/#/virus?SeqType_s=Nucleotide&VirusLineage_ss=Lyssavirus%20rabies,%20taxid:11292. Information on geography, country codes, colonization was incorporated via R packages: cepiigeogist, countrycode. All data used in this analysis are available on https://github.com/amholtz/GlobalRabies (https://doi.org/10.5281/zenodo.8047854)[33].

## Code availability

Analysis pipelines and ad-hoc Python3 and R scripts used for the analyses described above are available on https://github.com/amholtz/GlobalRabies (https://doi.org/10.5281/zenodo.8047854)[33].

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

## Acknowledgements

The authors thank S.L. Kosakovsky Pond for his help running HyPhy and interpreting the results. We would also like to thank Edward Holmes, Jake Faber and Joël Brehin for their helpful reading and critical comments in the review of the manuscript. G.B. acknowledges support from the Research Foundation - Flanders (Fonds voor Wetenschappelijk Onderzoek - Vlaanderen, FWO, Belgium; grant n° G098321N, G0E1420N) and from the Internal Funds KU Leuven (Grant No. C14/18/094). A.H. acknowledges École doctorale Frontières de l'Innovation en Recherche et Education-Programme Bettencourt. A.H is funded by the INCEPTION programme (Investissements d'Avenir grant ANR-16-CONV-0005). This work was funded by Institut Pasteur.

## Author contributions

A.H. and A.Z. conceived the idea and developed the theory. H.B. provided insight to rabies epidemiology. A.H. carried out the computational, phylogenetic and phylogeographic analysis with the supervision of A.Z. A.H. wrote the manuscript in consultation with A.Z. All authors provided critical feedback and helped shape the research and analysis. H.B. and G.B provided scientific and technical guidance and review of the paper.

## Competing interests

The authors declare no competing interests.
