## [Peer Review File · Nature Communications]

Integrating full and partial genome sequences to decipher the global spread of canine rabies virusREVIEWER COMMENTS

Reviewer #1 (Remarks to the Author):

The authors present a novel method to collate and interpret publicly available global sequence data for rabies virus. As with other endemic viral pathogens, whole genome data for RABV has increased over the last decade but there is a relatively underutilised wealth of information in partial genome data that can be used to explore wider historical and spatio-temporal patterns. However, it remains challenging to work with this data given a lack of standardisation in both the length and genome portion sequenced and it is often excluded or examined separately to WGS datasets. In my opinion a thoroughly validated approach to make use of such data has been a long time coming and this paper certainly addresses that issue. The authors fill a methodological gap to maximise insights from variable genetic data for RABV and many other long-studied pathogens. Simultaneously, they demonstrate the value of such collective data in phylogeographic reconstructions to pull out novel insights into rabies historical spread, such as the influence of colonisation, which have long been discussed but not quantitatively evidenced. In addition, and importantly, they assess the impact of widespread biases e.g. spatial under/over representation in genetic datasets and ways to deal with these biases, that can be used to inform future phylogenetic studies. Overall, this paper represents an important development in rabies (and other pathogens) phylogenetic studies and provides an updated and refined understanding of rabies virus global transmission dynamics. I am pleased to recommend its acceptance and publication, in consideration of some minor comments below.

1. I find it a little difficult to understand the concatenated sequence alignment and given that this is the foundation of the paper it would be helpful to have more detail

- o When you say alignments are concatenated you mean by row but also column-wise in some instances since some sequences originate from multi-gene or WGS? How are you linking the split gene sequences again (i.e. multi-gene sequences originating from the same sequence?)

- o I think some sort of visual to explain the alignment (supplementary would be fine) would be helpful here

- o Publishing the alignment on your git repo would be a useful resource

2. Clarification of ancestral character reconstruction. I think the fact that I am struggling to follow some of this highlights that it would benefit from some additional detail, see questions below

- o small point but ACR acronym isn't defined anywhere

- o With the ACR strategy, subtrees plus full tree, it's unclear to me how exactly you are doing this since the tree topology and internal nodes will differ to a certain degree. You aren't simply pruning an existing phylogeny but actually reconstructing it again, at least from what I can follow in the methods. Unless at this level of pruning you see little difference in internal node structure until quite superficially in phylogeny- in which case are you ignoring these more superficial nodes? And you say that you are finding identical cluster compositions- but they won't be identical since branches have been removed so you must be accepting nested matches (?). Can you explain more clearly?

- o You only apply the MPPA approach in the colonial ACR, why?

3. There is no discussion of the possible impact of recombination on this analysis. Although it's still a bit of a controversial topic in rabies community, there is still some evidence that it occurs. Do you think this could lead to some misinterpretation?

4. I think it's also worth noting that, while there may have been an increase in NCBI released sequences since 1970, there has also been a notable drop-off in submissions in the last 5-10 years that is not simply due to covid. Rather than a lack of sequencing I believe this is due to lack of data sharing (reasons ranging from in-country politics to embargos for publishing) and I wondered if you mention in discussion since open data sharing is an important aspect of genomic surveillance.

5. I think the reference Campbell et al 2022, Plos Path, should be mentioned and cited in your paper. This paper also highlights the problem of RABV seq data standardisation/availability and, in another way, attempts to interpret RABV diversity by making use of partial data alongside WGS.

6. Please add bootstrap values to the tree, at least at informative nodes

7. Can you providing some benchmarking stats as to how long trees take to build with different methods

8. A few details missing from methods such as threshold to label a branch a long-range/human-mediated transmission (in results), that you used a probability of 50% to consider a node resolved and taken forward for consensus aggregation (found this detail in a fig legend). Can you add such detail to methods section?

9. Canine sequences: I don't understand what you mean by "potential sequencing errors from gene-specific projects". Why would there be a gene specific sequencing error? Can you clarify

Reviewer #2 (Remarks to the Author):

In this manuscript, Holtz and colleagues develop a novel phylogenomic approach to utilize both partial and whole genome sequences of viruses consequential for human health, with Rabies virus (RABV) as their example. Using their approach, the authors are able to accurately date emergence events and lineage splits and estimate RABV geographic dispersal over hundreds of years by harnessing both partial and full genome sequences. This manuscript represents an important step in virus evolution and genomic epidemiology because many important viruses, most notably Dengue virus, have many partial genome sequences but not always complete genome sequences. This approach will allow researchers to add data to their genomic epidemiological analyses that previously would not have been included. This point is clearly made in Figure 1, that shows there is discordance between which gene sequences are generated and what region/host the sequences came from. The phylogenetic methods employed by the authors appear to be correct, and their conclusions are well supported by the data. Overall, this is a nice manuscript that I do not believe needs much additional work. Comments below:

Major Comments:

My only suggestion for this paper would be to expand upon how the sequence alignments were generated, as that is the novelty of the study. Its not entirely clear from reading the methods if the authors excluded samples that only had partial gene sequences. As well, on line 494, the authors note that gaps were introduced in non-sequenced locations of the genome- is this done by introducing "N" where there is no gene sequence for a particular sample? Finally, when all alignments were subsequently concatenated, do the authors mean that each sample, regardless of how much of the genome was actually sequenced, as a "full" genome in the final alignment? I think a cartoon of sorts could help readers visualize what the final alignments look like and can improve reproducibility of the method.

Reviewer #3 (Remarks to the Author):

The research article entitled "Deciphering the global spread of canine rabies virus in the modern era" by Holtz et al., is the most complete and compelling effort to reconstruct the contemporary evolutionary history of the dog rabies epizootic. The dog rabies epizootic was first documented in the Eshnunna code around the 23rd century B.C. However, it is believed that what became established as the dog-rabies virus (RABV) in pre-domesticated canines should have jumped from a common ancestor circulating in bats of the Old World long before the record found in the so-called first human civilization. It is likely that dog domestication ignited not only the massive proliferation of domesticated dogs, but also prompted the gradual human mediated spread of RABVs associated with dogs globally. Thus, dog maintained RABVs have been evolving through multiple anthropogenic introductions within domesticated dog populations and sympatric mesocarnivores species of the world for at least the last 20, 000 years. Over this long evolutionary

history, variants and subvariants of the RABV circulating in dog populations and mesocarnivores, across the urbanization gradient, have emerged and gone extinct. This apparently continued process of emergence and extinction across the globe has made it extremely difficult to track with high accuracy the date of the very origins and the first dissemination events within the dog rabies epizootic. The lack of historical RABV repositories or the availability of suitable tissues (to get RABV sequence data) from ancient rabid animals also complicates the overall reconstruction of the ancient episodes of this epizootic. Authors herein brilliantly tried to cover temporal gaps across the contemporary history of rabies not only by using concatenated-cistron whole genome sequences (WGS), but also by incorporating in to their analyses partial gene and partial genome sequences. Thus, authors compiled an unprecedented data set encompassing more than 14,000 sequences that cover more than 200 dissemination events across not neighboring countries. With this approach authors are not only breaking the paradigm that restricts accurate dating analyses to full-genome sequences, but also acknowledge the bias and noise inflicted by the inclusion of non-coding sequences in such analyses. Authors elegantly proved that their analysis could increase the accuracy for dating ancestral and recent nodes that help explain the dissemination dynamics and dates for the spread of the dog rabies epizootic across the world within the last 7 centuries. Authors were also able to depict the spread of dog-maintained RABVs in the context of major imperial invasions and most likely countries of origin. All main and supplementary figures are adequate and necessary. Author did not spare efforts to make their analyses more complete and the most robust possible by using sound novel methodologies for phylogenetic analysis and reconstruction. Thereby, authors attained the most coherent, complete spatiotemporal reconstruction of the contemporary history of the dog rabies epizootic. For the complexity, amount of sequence data collected, the time span comprised, and the geographic extension covered, this study has no precedent in the rabies field. Data visualization is both elegant and remarkable. Authors discussed major strengths and weaknesses in their analyses and clearly recognized limitations in their inferences. They made clear that their study does not resolve early (primal) dating and dissemination events within the evolutionary history of the dog rabies epizootic. However, it certainly sets the bases to improve dating and the reconstruction of earlier dog rabies dissemination events when RABVs from ancient rabid animals or humans may become available. After a thorough review I found no flaws or have comments for improvement worthy to mention. The research article is well written with a coherent, sound analytical support and an elegant data visualization component. The topic this research addresses is timely and highly relevant and shifts the paradigm that long restricted the use of partial genome and complete gene sequences together with whole genome sequences to reconstruct the contemporary evolutionary history of the dog-rabies epizootic. Authors thoroughly describe limitations of their analysis rendering an honest account of the contemporary spatiotemporal dissemination dynamics of the global dog rabies epizootic.

Reviewers Comments:

REVIEWER COMMENTS

Reviewer #1 (Remarks to the Author):

The authors present a novel method to collate and interpret publicly available global sequence data for rabies virus. As with other endemic viral pathogens, whole genome data for RABV has increased over the last decade but there is a relatively underutilised wealth of information in partial genome data that can be used to explore wider historical and spatio-temporal patterns. However, it remains challenging to work with this data given a lack of standardisation in both the length and genome portion sequenced and it is often excluded or examined separately to WGS datasets. In my opinion a thoroughly validated approach to make use of such data has been a long time coming and this paper certainly addresses that issue. The authors fill a methodological gap to maximise insights from variable genetic data for RABV and many other long-studied pathogens. Simultaneously, they demonstrate the value of such collective data in phylogeographic reconstructions to pull out novel insights into rabies historical spread, such as the influence of colonisation, which have long been discussed but not quantitatively evidenced. In addition, and importantly, they assess the impact of widespread biases e.g. spatial under/over representation in genetic datasets and ways to deal with these biases, that can be used to inform future phylogenetic studies. Overall, this paper represents an important development in rabies (and other pathogens) phylogenetic studies and provides an updated and refined understanding of rabies virus global transmission dynamics. I am pleased to recommend its acceptance and publication, in consideration of some minor comments below.

Response: We thank the Reviewer for this positive assessment of our work.

- 1. I find it a little difficult to understand the concatenated sequence alignment and given that this is the foundation of the paper it would be helpful to have more detail**
 - o When you say alignments are concatenated you mean by row but also column-wise in some instances since some sequences originate from multi-gene or WGS? How are you linking the split gene sequences again (i.e. multi-gene sequences originating from the same sequence?)**
 - o I think some sort of visual to explain the alignment (supplementary would be fine) would be helpful here**

Response: We thank the Reviewer for this suggestion and have created an additional figure to clarify this issue (Supplementary Figure 2). We show the figure here for your convenience.

RABV Gene Concatenation-Multiple Sequence Alignment Scheme

Supplementary Figure 2. Cartoon demonstration on the method of partial and WGS sequence concatenation. An initial global alignment was used to order the sequences in gene order. Sequences were then sorted into gene-specific fasta files with sequences cut at the start and stop codons of the corresponding gene. A custom script was used to identify sequences with more than 200 nucleotides in each gene's coding region. The gene-specific fasta files were aligned and concatenated to form a global multiple sequence alignment with a total length of 10,860 nucleotides. The illustration of the RABV genome is shown at the top. Numbers inside each gene area define the positions from the reference genome where the gene-specific sequences were cut. The final resulting multiple sequence alignment is shown in the box on the bottom right. Corresponding file names on the GitHub page (https://github.com/amholtz/GlobalRabies/tree/main/data/sequence_alignments/gene_specific_analysis) are used.

As can be seen in the figure above, our concatenation approach glues together the sequences in a column-wise manner as part of the final step in the creation of the sequence alignment. The P gene is glued directly downstream of the N gene, and the M gene is glued directly downstream of the P gene. This process continues until we finally glue the L gene downstream of the G gene. The concatenation column-wise is void of all noncoding regions, which were removed from the gene-specific alignments at an earlier step.

o Publishing the alignment on your git repo would be a useful resource

The large alignment files (>100MB) have been added to the GitHub repository, inside the data folder, using Git Large File Storage (LFS).

2. Clarification of ancestral character reconstruction. In think the fact that I am struggling to follow some of this highlights that it would benefit from some additional detail, see questions below

- o small point but ACR acronym isn't defined anywhere

Response: We thank the Reviewer for pointing this out and have now defined this in the Results section of the text.

- o With the ACR strategy, subtrees plus full tree, it's unclear to me how exactly you are doing this since the tree topology and internal nodes will differ to a certain degree. You aren't simply pruning an existing phylogeny but actually reconstructing it again, at least from what I can follow in the methods. Unless at this level of pruning you see little difference in internal node structure until quite superficially in phylogeny- in which case are you ignoring these more superficial nodes? And you say that you are finding identical cluster compositions- but they won't be identical since branches have been removed so you must be accepting nested matches (?). Can you explain more clearly?

Response:

Indeed, we reconstructed the phylogenetic trees over again from the subsampling, instead of simply pruning. This is because we wanted to use the reconstructions on small trees as a way to test the confidence of the larger tree. If we simply pruned the trees, the topology would be the same. To compare the trees, we wrote an R script, ACR_Sub_comparison.R, which searches for similar nodes (most recent common ancestors of the same tips) among different trees. The visualization below, which has been newly added as Supp. Figure 6, illustrates how this concept operates using a small example.

Supplementary Figure 6. Mock example of internal node comparison. The Full Tree (on the left) contains nine tips. Tips removed during subsampling are marked with an X. Subsample 1 (on the right) contains 6 tips. Subsample 1 has been completely newly reconstructed using IQ-TREE2 starting from the

subsampled sequence alignment. Notice that the topologies between the Full Tree (FT) and Subsample 1 are not identical (the light blue and green nodes have swapped). The goal is to find the node in the Full Tree which is the most recent common ancestor of all the descendant tips of a node of Subsample 1. In this example, node 4 in Subsample 1 can be compared to node 4 in the Full Tree. Node 3 can be compared to node 4 as well. Node 2 in Subsample 1 can be compared to node 3 in the Full Tree. Node 1 in Subsample 1 can be compared to Node 1 in the Full Tree. If you now compare the ACR, you can see that the full tree nodes that share the same ACR are 1 and 3. This is the idea of the node comparison script which was applied on a much larger scale.

o You only apply the MPPA approach in the colonial ACR, why?

Response: The MPPA approach was used for both the colonial and regional ACR. This approach is more thorough since it uses the Brier score to select a subset of probable states for each node, which helps minimize prediction error. For the ACR on the country level, however, we needed one distinct state per node to make the consensus tree from the subsampled and full trees. Our script, ACR_Sub_comparsion.R, created the consensus tree by defining the ancestral character as a character with a marginal posterior probability of 50% or more.

3. There is no discussion of the possible impact of recombination on this analysis. Although it's still a bit of a controversial topic in the rabies community, there is still some evidence that it occurs. Do you think this could lead to some misinterpretation?

Response: In recent years, there have been some studies investigating recombination in RABV sequences. An overview of the topic was reviewed in Deviakin (2018), where the authors looked for recombination events in the genome using modern tools and softwares. Their main conclusion is that the overall impact of recombination in RABV evolution is low, and if it does play a role it would be in bat-mediated RABV. Within the context of RABV infection dynamics, this makes sense. In order to have recombination events, you need coinfection of two different types of the same virus. These types of events in canine-mediated RABV are rare due to the quick fatality of an infected individual and the infection/transmission rate. There is evidence, however, in bat communities that RABV infection is not fatal, which would allow for coinfection of multiple RABV strains. In our study, we only focus on canine-mediated rabies. Furthermore, as we and others have shown, canine-mediated rabies has a distinct geographical distribution of genetic variants. Typically, only one genetic variant is found in a specific host population in a given area. Consequently, recombination is likely to happen among closely related viruses, so any recombination events would likely be invisible on a global large-scale phylogenetic analysis like ours. To conclude, even if recombination were to occur in canine-mediate RABV, we do not believe its impact would be visible on a global phylogenetic tree. This is now discussed in a distinct paragraph in the Discussion section.

4. I think it's also worth noting that, while there may have been an increase in NCBI released sequences since 1970, there has also been a notable drop-off in submissions in the last 5-10 years that is not simply due to covid. Rather than a lack of sequencing I believe this is due to lack of data sharing (reasons ranging from in-country politics to embargos for publishing) and I wondered if you mention in discussion since open data sharing is an important aspect of genomic surveillance.

Response: Thank you for bringing light to this important point. We have included it now in the discussion.

5. I think the reference Campbell et al 2022, Plos Path, should be mentioned and cited in your paper. This paper also highlights the problem of RABV seq data standardisation/availability and, in another way, attempts to interpret RABV diversity by making use of partial data alongside WGS.

Response: We have now mentioned and cited this paper in the section of the discussion where we write about the bias and limitations.

6. Please add bootstrap values to the tree, at least at informative nodes

Response: We have added bootstrap values to the global tree reconstruction (Supp. Figure 2; now 3) for the major clades of interest (bats, raccoons-skunks, indian, asian, africa-2, arctic, africa-3, cosmopolitan). We have also added bootstrap values to Figure 3. Figure 3 is a consensus tree from 5 subsampled trees reconstructed in IQ-TREE2 and FastTree. The bootstrap values displayed are the lowest values from the subsampled tree reconstructions estimated using IQ-TREE2. All but four out of the bootstrap values for the 44 clade-defining nodes in all 5 subsample tree reconstructions are 1 (those four are > 0.85).

7. Can you providing some benchmarking stats as to how long trees take to build with different methods

FastTree (n=14,752 sequences) on 32 threads: 4533.09 seconds = ~1h:15min
FastTree (n=10,209 sequences) on 32 threads: 2851.89 seconds = ~47min
Subsample1 IQ-TREE2 (n= 5,551 sequences) on 8 threads = 164h:28min:49s

These benchmark stats have now been added to the Methods section of the manuscript.

8. A few details missing from methods such as threshold to label a branch a long-range/human-mediated transmission (in results), that you used to consider a node resolved and taken forward for consensus aggregation (found this detail in a fig legend). Can you add such detail to methods section?

Response: The definition of a resolved node is included in Supplemental Methods 2, but we have now added both of these points into the main method section of the manuscript as well.

9. Canine sequences: I don't understand what you mean by "potential sequencing errors from gene-specific projects". Why would there be a gene specific sequencing error? Can you clarify.

Response: We agree, this is confusing. This phrase was removed and replaced with the following: *"The evolutionary rate was estimated using whole genome sequences exclusively for robustness"*.

Reviewer #2 (Remarks to the Author):

In this manuscript, Holtz and colleagues develop a novel phylogenomic approach to utilize both partial and whole genome sequences of viruses consequential for human health, with Rabies virus (RABV) as their example. Using their approach, the authors are able to accurately date emergence events and lineage splits and estimate RABV geographic dispersal over hundreds of years by harnessing both partial and full genome sequences. This manuscript represents an important step in virus evolution and genomic epidemiology because many important viruses, most notably Dengue virus, have many partial genome sequences but not always complete genome sequences. This approach will allow researchers to add data to their genomic epidemiological analyses that previously would not have been included. This point is clearly made in Figure 1, that shows there is discordance between which gene sequences are generated and what region/host the sequences came from. The phylogenetic methods employed by the authors appear to be correct, and their conclusions are well supported by the data. Overall, this is a nice manuscript that I do not believe needs much additional work. Comments below:

Response: We thank the Reviewer for this positive assessment of our work.

Major Comments:

My only suggestion for this paper would be to expand upon how the sequence alignments were generated, as that is the novelty of the study. Its not entirely clear from reading the methods if the authors excluded samples that only had partial gene sequences.

Response: We thank the Reviewer for pointing this out and have now expanded the explanation in the main text and included Supplementary Figure 2, which should clear this confusion (see answer to reviewer 1's comments).

As well, on line 494, the authors note that gaps were introduced in non-sequenced locations of the genome- is this done by introducing "N" where there is no gene sequence for a particular sample?

Response: Instead of introducing 'N's, we introduced gaps (-) in the non-sequenced gene regions. Standard maximum-likelihood tree reconstruction methods with GTR-like models treat gaps and Ns in the same way (considering them as any possible character, i.e. A, C, G or T). This is now shown in the new Supplementary Figure 2 and made more clear in the Methods section.

Finally, when all alignments were subsequently concatenated, do the authors mean that each sample, regardless of how much of the genome was actually sequenced, as a “full” genome in the final alignment? I think a cartoon of sorts could help readers visualize what the final alignments look like and can improve reproducibility of the method.

Response: Yes, that is correct. In the final alignment, each sequence has 10860 sites in the sequence alignment (see answer to reviewer 1's comments). IDs that are original whole genome sequences will have nucleotides for all 10860 sites, whereas partial sequences will have nucleotides in the sequenced region, and then gaps for the remaining. For example, ID MW574110 is a sequence from the N gene. In the final alignment ID MW574110 will have the nucleotides from the coding region of the N gene sequence, plus gaps for the remaining part of the genome.

We have created a new visualization, Supplementary Figure 2, which should help explain the construction of the final alignment.

Reviewer #3 (Remarks to the Author):

The research article entitled “Deciphering the global spread of canine rabies virus in the modern era” by Holtz et al., is the most complete and compelling effort to reconstruct the contemporary evolutionary history of the dog rabies epizootic. The dog rabies epizootic was first documented in the Eshnunna code around the 23rd century B.C. However, it is believed that what became established as the dog-rabies virus (RABV) in pre-domesticated canines should have jumped from a common ancestor circulating in bats of the Old World long before the record found in the so-called first human civilization. It is likely that dog domestication ignited not only the massive proliferation of domesticated dogs, but also prompted the gradual human mediated spread of RABVs associated with dogs globally. Thus, dog maintained RABVs have been evolving through multiple anthropogenic introductions within domesticated dog populations and sympatric mesocarnivores species of the world for at least the last 20, 000 years. Over this long evolutionary history, variants and subvariants of the RABV circulating in dog populations and mesocarnivores, across the urbanization gradient, have emerge and gone extinct. This apparently continued process of emergence and extinction across the globe has made extremely difficult to track with high accuracy the date of the very origins and the first

dissemination events within the dog rabies epizootic. The lack of historical RABV repositories or the availability of suitable tissues (to get RABV sequence data) from ancient rabid animals also complicates the overall reconstruction of the ancient episodes of this epizootic. Authors herein brilliantly tried to cover temporal gaps across the contemporary history of rabies not only by using concatenated-cistron whole genome sequences (WGS), but also by incorporating in to their analyses partial gen and partial genome sequences. Thus, authors compiled an unprecedented data set encompassing more than 14, 000 sequences that cover more than 200 dissemination events across not neighboring countries. With this approach authors are not only breaking the paradigm that restricts accurate dating analyses to full-genome sequences, but also acknowledge the bias and noise inflicted by the inclusion of non-coding sequences in such analyses. Authors elegantly proved that their analysis could increased the accuracy for dating ancestral and recent nodes that help explain the dissemination dynamics and dates for the spread of the dog rabies epizootic across the world within the last 7 centuries. Authors were also able to depict the spread of dog-maintained RABVs in the context of major imperial invasions and most likely countries of origin. All main and supplementary figures are adequate and necessary. Author did not spare efforts to make their analyses more complete and the robust possible by using sounded novel methodologies for phylogetic analysis and reconstruction. Thereby, authors attained the most coherent, complete spatiotemporal reconstruction of the contemporary history of the dog rabies epizootic. For the complexity, amount of sequence data collected, the time span comprised, and the geographic extension covered, this study has no precedent in the rabies field. Data visualization is both elegant and remarkable.

Authors discussed major strengths and weaknesses in their analyses and clearly recognized limitations in their inferences. They made clear that their study does not resolve early (primal) dating and disseminations events within the evolutionary history of the dog rabies epizootic. However, it certainly set the bases to improve dating and the reconstruction of earlier dog rabies dissemination events when RABVs from ancient rabid animals or humans may become available.

After a thorough review I found no flaws or have comments for improvement worthy to mention. The research article is well written with a coherent, sounded analytical support and an elegant data visualization component. The topic this research addresses is timely and highly relevant and shifts the paradigm that long restricted the use of partial genome and complete gene sequences together with whole genome sequences to reconstruct the contemporary evolutionary history of the dog-rabies epizootic. Authors thoroughly describe limitations of their analysis rendering and honest account of the contemporary spatiotemporal dissemination dynamics of the global dog rabies epizootic.

Response: We thank the Reviewer for this positive assessment of our work.

REVIEWERS' COMMENTS

Reviewer #1 (Remarks to the Author):

The authors have addressed the reviewer's comments and I am happy to recommend for publication.